# Diagnostic Performance of Ring Aperture Retro Mode Imaging for Detecting Pigment Migration in Age-Related Macular Degeneration

**DOI:** 10.3390/diagnostics16010042

**Published:** 2025-12-23

**Authors:** Thomas Desmettre, Gerardo Ledesma-Gil, Michel Paques

**Affiliations:** 1Centre de Rétine Médicale, 187 Rue de Menin, 59520 Marquette-Lez-Lille, France; 2Department of Ophthalmology, University of Kansas School of Medicine, Prairie Village, KS 66208, USA; 3Retina Department, Institute of Ophthalmology, Fundación Conde de Valenciana, Mexico City 06800, Mexico; gerardo.ledesma.md@gmail.com; 4Paris Eye Imaging Group, Clinical Investigation Center 1423, Quinze-Vingts Hospital, National Institute of Health and Medical Research-Directorate of Hospitalization and Healthcare Organization, 75012 Paris, France; michel.paques@protonmail.com

**Keywords:** age-related macular degeneration, geographic atrophy, retinal imaging, diagnostic performance, en face OCT, retro mode imaging, pigment migration, hyperreflective foci, thickened RPE, refractile drusen

## Abstract

**Background/Objectives**: Pigment migration is a key biomarker of progression in age-related macular degeneration (AMD). This study assessed the diagnostic performance of ring aperture Retro mode (RAR) imaging for detecting pigment migration and compared its performance with established multimodal imaging techniques. **Methods**: This retrospective study included 80 eyes from 61 consecutive patients with AMD who underwent multimodal imaging with color fundus images (CFIs), fundus autofluorescence (FAF), RAR imaging (Mirante, NIDEK), and en face optical coherence tomography (OCT) with B-scans (Cirrus HD-OCT 5000, Zeiss). Two independent retina specialists graded the AMD stage and the presence of pigment migration across modalities. Sensitivity and positive predictive value (PPV) of RAR were calculated using en face OCT as the reference standard. **Results**: RAR demonstrated high diagnostic performance, with a sensitivity of 94.7% and a PPV of 93.4% relative to en face OCT. RAR frequently identified pigment migration that was not visible on CFI or FAF, particularly in early AMD and in eyes with media opacity. Distinct morphologic patterns—including hyperreflective foci, thickened retinal pigment epithelium, refractile drusen, and cuticular drusen—were consistently identifiable on RAR. In four eyes with geographic atrophy, RAR detected perifoveal pigment redistribution at least six months before foveal involvement was confirmed by OCT and FAF. **Conclusions**: RAR imaging is a rapid, sensitive, and clinically practical technique for detecting pigment migration in AMD. By complementing en face OCT and enhancing visualization in cases where standard imaging is limited, RAR may strengthen early disease surveillance, support prognostic assessment, and improve multimodal diagnostic workflows in routine practice.

## 1. Introduction

Pigment migration in age-related macular degeneration (AMD) is an important clinical finding, reflecting retinal pigment epithelium (RPE) dysfunction and signaling risk of progression toward geographic atrophy (GA) and vision loss [1]. Along with drusen, pigmentary changes are a central criterion for upgrading disease classification from early to intermediate AMD (iAMD) [1]. According to the Beckman/AREDS2 consensus classification, iAMD is defined by the presence of large drusen (>125 µm) and/or any AMD-related pigmentary abnormalities in the absence of GA or neovascularization [2].

Pigmentary abnormalities encompass both hyperpigmentation—reflecting RPE migration, hyperreflective foci (HRF), or thickened RPE (tRPE)—and hypopigmentation, which may correspond to early RPE attenuation, nascent GA, or emerging transmission defects on OCT [3]. These changes occur within the tightly coupled RPE–photoreceptor unit. Although RPE dysfunction can compromise photoreceptor survival [4,5], recent OCT-based analyses of AMD progression indicate that photoreceptor alterations with ellipsoid-zone attenuation may also extend beyond or precede detectable RPE loss in early GA, highlighting a bidirectional and stage-dependent relationship between these layers [6].

Multimodal imaging has improved the detection of pigment migration. Color fundus images (CFIs) may reveal visible pigment clumping, while structural and en face OCT localize abnormalities as HRF or tRPE [3]. Hyporeflective clumps (HRCs) are another lesion detectable with adaptive optics [7,8]. These abnormalities may occur over drusen, at atrophic margins, or on the surface of pigment epithelial detachments (PEDs). En face OCT can further delineate pigment migration associated with refractile drusen or cuticular drusen [9,10,11].

Fundus autofluorescence (FAF) demonstrates pigment redistribution through characteristic patterns [12]; the main signal sources for FAF are RPE lipofuscin and melanolipofuscin [13]. Because the relative contribution of these fluorophores depends on excitation wavelength, conventional visible-light FAF predominantly reflects lipofuscin. Melanolipofuscin, however, exhibits broader emission properties and can generate near-infrared autofluorescence under long-wavelength excitation [14].

Infrared retroillumination (“Retro mode”) was first introduced on the NIDEK F-10 platform and later integrated into the Mirante^®^ multimodal device [15,16]. The system offers three illumination apertures: deviated left (DL), deviated right (DR), and ring aperture (RA). DL and DR configurations create pseudo-three-dimensional shading that accentuates drusen contours, pigment epithelial detachments, and atrophic margins, making them widely used for structural assessment in AMD [17,18,19,20].

In contrast, the ring aperture configuration (RAR) eliminates lateral shadowing and produces a “dark-field” transillumination pattern. Although less visually striking than DL/DR imaging, RAR preferentially highlights small variations in pigment density. When acquisition parameters—particularly flash intensity—are carefully optimized, RAR may reveal subtle pigment redistribution that is difficult to detect on CFIs or FAF, especially near the fovea, where macular pigment absorption complicates interpretation.

Despite this potential, the diagnostic value of RAR for detecting pigment migration in AMD has not, to the best of our knowledge, been systematically evaluated. The present study aimed to assess the performance of RAR for detecting pigment migration across different AMD stages and to compare its yield with that of color fundus imaging, FAF, and en face OCT in a real-world, high-volume clinical setting.

## 2. Materials and Methods

### 2.1. Study Design

This retrospective observational study was conducted at the Centre de Rétine Médicale, a tertiary referral center for retinal diseases. The study complied with the principles of the Declaration of Helsinki. Written informed consent was obtained from all participants, authorizing the use of anonymized clinical data for research purposes. Ethical approval was obtained from the Ethics Committee of the French Society of Ophthalmology (SFO) (IRB 00008855 SFO IRB#1).

### 2.2. Patients

Inclusion criteria were consecutive patients examined between May 2023 and February 2025 with a diagnosis of age-related macular degeneration (AMD) and evidence of pigment migration detected on at least one imaging modality. Pigment migration was independently identified by two retina specialists (TD, GL-G).

Exclusion criteria were insufficient image quality or the absence of one or more required imaging modalities.

Based on multimodal imaging characteristics, eyes were classified as intermediate AMD (iAMD), geographic atrophy (GA), or neovascular AMD (nAMD). Eyes presenting both GA and nAMD were classified in the nAMD group.

Best-corrected visual acuity (BCVA) was recorded to provide clinical context but was not used as an outcome measure, as pigment migration is not always directly correlated with central visual function, particularly in the presence of GA or media opacity. Refraction was not included in the analytical models because mild ametropia does not influence the detection of pigment migration; refractive status is reported descriptively in the Results section for completeness. The primary objective of the study was to evaluate imaging detectability rather than functional impact.

### 2.3. Imaging Protocol

All imaging data were anonymized prior to analysis. Examinations were performed after pharmacologic pupil dilation. All images were acquired by a single experienced retina specialist (TD), ensuring consistent optimization of acquisition parameters across modalities. This approach ensured optimal image quality for methodological evaluation; generalizability to broader acquisition settings warrants future validation.

Multimodal imaging included color fundus images (CFIs), fundus autofluorescence (FAF), and Retro mode imaging (deviated left [DL], deviated right [DR], and ring aperture [RA]) acquired with the Mirante^®^ platform (NIDEK, Gamagori, Japan; 89° field of view), as well as spectral-domain optical coherence tomography (OCT) using the Cirrus HD-OCT 5000 (Carl Zeiss Meditec, Dublin, CA, USA; axial resolution ~5 µm; acquisition speed 27,000 A-scans/second).

CFIs were obtained with carefully adjusted flash intensity to avoid overexposure, particularly in eyes with GA. Retro mode imaging included DL, DR, and RA acquisitions; however, only RA (RAR) images were analyzed in the present study. Flash intensity was progressively reduced to balance contrast and allow visualization of both choroidal vessels and pigment migration. FAF was performed using green excitation (532 nm), minimizing glare and reducing macular pigment absorption.

### 2.4. En Face OCT Acquisition

En face OCT data were derived from 6 × 6 mm scans centered on the fovea. Two Cirrus preset slabs were analyzed across all eyes:
Retinal IS/OS slab

Used to detect intraretinal hyperreflective foci (HRF), which appear as bright focal dots at the photoreceptor integrity line.

2.Sub-RPE/choroidal slab

Located beneath the RPE/Bruch’s complex, used to evaluate transmission defects.

Hypotransmission defects (hypoTD): dark foci caused by signal blockage from thickened RPE (tRPE) or large HRF.Hypertransmission defects (hyperTD): bright patches corresponding to RPE attenuation or early atrophy.

For all en face abnormalities, the corresponding B-scans were reviewed to determine whether the lesion represented HRF, tRPE, or RPE loss. This approach follows established en face slab-dependent behavior described in the literature [3,11]. Although customized slabs can reveal additional lesion characteristics, preset slabs were intentionally used to ensure reproducibility and clinical applicability across routine practice.

### 2.5. Image Analysis

All images were independently reviewed by two retina specialists (TD and GL-G). The graders did not undergo a joint calibration session and independently evaluated images using predefined criteria. Images from different imaging modalities were reviewed in separate reading sessions to minimize cross-modality recall bias. Both graders were masked to each other’s findings. After an independent assessment, discrepancies were resolved by consensus. For each eye, graders classified the AMD stage (intermediate AMD, neovascular AMD, or geographic atrophy) and scored the presence of pigment migration for each imaging modality as binary outcomes (“0” = absent; “1” = present).

### 2.6. Statistical Analysis

Intergrader agreement for binary variables (presence of pigment migration on each imaging modality) was assessed using Cohen’s κ coefficient. Agreement for AMD stage classification was evaluated using weighted κ. Kappa values were interpreted according to Landis and Koch’s criteria. Ninety-five percent confidence intervals (95% CIs) were calculated for all κ estimates.

Diagnostic performance of CFIs, FAF, and RAR was evaluated using en face OCT, corroborated by B-scan OCT as the reference standard. From 2 × 2 contingency tables, sensitivity, specificity, positive predictive value (PPV), negative predictive value (NPV, when applicable), and the F1 score were calculated. The F1 score, defined as the harmonic mean of sensitivity and PPV, was included to provide a balanced measure of detection performance in a dataset enriched for positive findings.

Because both eyes from some patients were included, statistical uncertainty was estimated using a patient-level cluster bootstrap approach. Patients were resampled with replacement, retaining all corresponding eyes within each resample, thereby accounting for within-subject correlation and avoiding underestimation of variance. Ninety-five percent confidence intervals for sensitivity, PPV, F1 score, and κ statistics were derived from bootstrap distributions.

Continuous variables are reported as mean ± standard deviation, and categorical variables as counts and percentages. Given the retrospective design and enrichment for pigment migration, specificity and NPV estimates are interpreted cautiously and reported primarily for completeness. Statistical analyses were performed using standard spreadsheet tools, with custom resampling procedures implemented for bootstrap-based confidence interval estimation. A two-sided significance threshold of *p* < 0.05 was applied where appropriate.

## 3. Results

A total of 80 eyes from 61 patients were included. The mean age was 79.1 ± 9.8 years, and 39 patients (64%) were women. Based on multimodal imaging, 36 eyes were classified as intermediate AMD (iAMD), 23 as geographic atrophy (GA), and 21 as neovascular AMD (nAMD); eyes presenting both GA and nAMD were grouped within the nAMD category.

The mean best-corrected visual acuity (BCVA) was 0.59 ± 0.26 (decimal units; approximately 20/34 Snellen; mean logMAR 0.28 ± 0.24). Available refraction data indicated mild ametropia, with spherical equivalents ranging approximately from −3.00 D to +2.00 D; no eyes with high myopia or high hyperopia were included. Visual acuity was recorded for clinical characterization but was not used as an outcome measure.

For all eyes, en face OCT findings were interpreted by correlating abnormalities observed on both the ellipsoid zone and sub-RPE/choroidal slabs with the corresponding B-scans to differentiate hyperreflective foci (HRF), thickened RPE (tRPE), and transmission defects. FAF findings were analyzed descriptively, as signal interpretation varies with macular pigment absorption, fluorophore composition, and disease stage, limiting its suitability as a binary comparator.

### 3.1. Intergrader Agreement

Intergrader agreement was excellent for AMD stage classification (weighted κ = 0.93). Agreement for detection of pigment migration was similarly high across imaging modalities, including CFIs (κ = 0.92), FAF (κ = 0.83), en face OCT (κ = 0.90), and RAR (κ = 1.00). For all κ statistics, 95% confidence intervals were estimated using patient-level cluster bootstrap resampling to account for intra-subject correlation due to the inclusion of both eyes from some participants.

### 3.2. Diagnostic Performance

Using en face OCT, corroborated by B-scans as the reference standard, RAR demonstrated high agreement (κ = 0.71; 95% CI: 0.53–0.89), with a sensitivity of 94.7% (95% CI: 86.9–98.5%) and a positive predictive value (PPV) of 93.4% (95% CI: 85.3–97.6%). Agreement between RAR and CFIs was moderate (κ = 0.56), with a sensitivity of 91.5% (95% CI: 80.1–96.6%) and a PPV of 56.6% (95% CI: 45.4–67.1%). Most RAR “false positives” relative to CFIs corresponded to pigment migration confirmed on en face OCT, suggesting that these lesions were more likely missed on color fundus photographs rather than representing spurious RAR findings.

Because the cohort was enriched for eyes with pigment migration, and en face OCT identified pigment migration in most eyes, true negatives were infrequent, resulting in non-informative estimates of specificity and negative predictive value. Contingency tables are shown in Table 1 and performance metrics in Table 2.

All confidence intervals for sensitivity, PPV, F1 score, and κ statistics were derived using a cluster bootstrap approach with resampling at the patient level to account for intra-subject correlation.

### 3.3. Morphologic Patterns

Distinct morphologic patterns of pigment migration were identified on RAR and correlated with OCT findings (Table 3). On RAR, pigment migration corresponding to HRF on B-scan OCT appeared as thin dark contrast dots or mid-sized dark contrast halos (Figure 1 and Figure 2). Areas of thickened RPE (tRPE) were visualized as broader dark contrast halos (Figure 3). Refractile drusen, showing a pyramidal contour with central hypertransmission on OCT B-scans, appeared as ring-shaped dark halos with a bright central core on RAR (Figure 4). Cuticular drusen were visible as small, clustered lesions with central dark halos and peripheral light rings (Figure 5).

Compared with CFIs and FAF, RAR frequently provided clearer visualization of pigment migration, particularly in the presence of media opacities or near the fovea. In four eyes with GA, subtle pigment migration was detected on RAR, preceded by foveal involvement by at least six months, later confirmed by OCT and FAF. In one eye with iAMD and cuticular drusen, FAF provided superior visualization, underscoring the complementary value of multimodal imaging.

Distinct morphologic patterns of pigment migration were identified on RAR when compared with OCT (Table 3).

## 4. Discussion

In this retrospective study, ring aperture Retro mode (RAR) imaging proved to be a valuable modality for detecting pigment migration across different stages of age-related macular degeneration (AMD). Its sensitivity was comparable to en face OCT, the current structural reference method, and superior to color fundus images (CFIs) and fundus autofluorescence (FAF) in selected situations, particularly for early or subtle pigmentary changes. The combination of rapid acquisition and intuitive interpretation supports the integration of RAR into multimodal imaging workflows, especially in high-volume clinical settings.

Pigmentary changes are recognized biomarkers of AMD progression, reflecting retinal pigment epithelium (RPE) dysfunction and an increased risk of conversion to geographic atrophy (GA) or neovascular disease [1,4,5]. These lesions are often subtle and difficult to detect on CFIs or FAF, particularly in the foveal region, where macular pigment absorption can obscure abnormalities [23]. Although green autofluorescence reduces macular pigment absorption compared with blue-light FAF [13,24], FAF signals in GA arise from multiple mechanisms beyond lipofuscin and melanolipofuscin accumulation [13,24]. In contrast, RAR relies primarily on variations in pigment density and uses infrared illumination, which bypasses macular pigment interference. As a result, RAR consistently revealed pigment migration underestimated or missed by CFIs and FAF, particularly near the fovea or in the presence of media opacities (Figure 1 and Figure 5; Table 1 and Table 2).

Distinct morphologic patterns of pigment migration identified on RAR provide further insight into the underlying pathophysiology. Hyperreflective foci (HRF), well-established OCT biomarkers of AMD progression [25,26,27], appeared on RAR as thin dark contrast dots or mid-sized dark contrast halos (Figure 1 and Figure 2). Thickened RPE (tRPE), associated with GA progression and basal laminar deposits [3,28,29,30], was visualized as broader dark contrast halos (Figure 3). Refractile drusen, typically pyramidal on OCT with central hypertransmission [10,11], appeared on RAR as ring-shaped dark contrast halos with a bright central core (Figure 4). Cuticular drusen were visualized on RAR as small, clustered lesions with central dark halos and peripheral light rings (Figure 5). These patterns, summarized in Table 3, align with histopathologic and multimodal imaging studies highlighting the complexity of RPE–photoreceptor interactions [4,10,31,32].

RAR also provided clinically meaningful information during follow-up. In four eyes with GA, pigment migration detected on RAR preceded foveal involvement by at least six months, later confirmed by en face OCT and FAF (Figure 5). These observations suggest that RAR may help identify early pigment redistribution with potential prognostic relevance for central atrophy progression. Similarly, in two cases of neovascular AMD with persistent shallow subretinal detachment, pigment migration was obscured on CFIs and FAF but remained visible on RAR and OCT. Chronic subretinal fluid can generate hyperautofluorescent deposits that mask pigment redistribution on FAF [33], whereas RAR retained diagnostic utility in this context.

Near-infrared autofluorescence (NIR-AF) imaging has emerged as a complementary approach for assessing pigmentary alterations in AMD, particularly through its sensitivity to melanolipofuscin and melanin-related fluorophores [13,14]. While NIR-AF provides direct autofluorescence signals related to pigment composition, RAR relies on infrared transillumination and contrast modulation induced by pigment density and scattering properties rather than intrinsic fluorescence. Consequently, RAR may highlight pigment redistribution even in regions where autofluorescence signals are complex, attenuated, or confounded by overlapping fluorophores. These distinct physical principles suggest that RAR and NIR-AF capture complementary aspects of pigment remodeling. Prospective longitudinal studies integrating RAR with OCT and near-infrared autofluorescence may further refine the role of pigment redistribution as a biomarker of AMD progression.

This study has limitations, including its retrospective design and modest sample size. Although graders were not masked to clinical diagnosis, image interpretation was performed independently and masked to the other grader’s results, with high intergrader agreement supporting reliability. All images were acquired by a single experienced operator, which ensured standardized acquisition and optimized image quality but may limit generalizability to routine clinical settings. In particular, RAR image quality depends on careful adjustment of illumination parameters, suggesting a learning curve similar to that previously described for deviated-aperture (DL/DR) Retro mode imaging.

While en face OCT remains the structural reference standard, its interpretation requires systematic evaluation of multiple slabs and corresponding B-scans, increasing interpretive complexity. In addition, en face OCT is subject to slab-dependent artifacts—including projection effects, mirror artifacts, and shadowing from overlying structures—that can influence lesion appearance depending on slab depth and thickness. In the present study, these effects were mitigated through consistent slab selection and mandatory B-scan correlation; nonetheless, they highlight intrinsic limitations of slab-based OCT analysis when assessing subtle pigment redistribution.

Finally, although distinct morphologic patterns were identified, this study was not designed to validate RAR as a definitive discriminator among HRF, tRPE, and other lesion subtypes. Nonetheless, the consistency of findings across imaging modalities supports the clinical utility of RAR as part of a streamlined multimodal approach for AMD.

In the future, longitudinal studies integrating RAR and other imaging modalities may further refine the role of pigment redistribution as a biomarker of AMD progression.

## 5. Conclusions

Ring aperture Retro mode imaging provides a rapid and sensitive method for detecting pigment migration in age-related macular degeneration, with diagnostic performance comparable to en face OCT and superior detection compared with color fundus imaging and fundus autofluorescence in selected situations. By enhancing visualization of subtle pigment redistribution—particularly in early disease stages, near the fovea, or in the presence of media opacity—RAR may offer added clinical value for routine monitoring and prognostic assessment. As a complement to OCT-based structural analysis, RAR represents a practical adjunct in multimodal imaging strategies for AMD in real-world clinical practice.

## Figures and Tables

**Figure 1 diagnostics-16-00042-f001:**
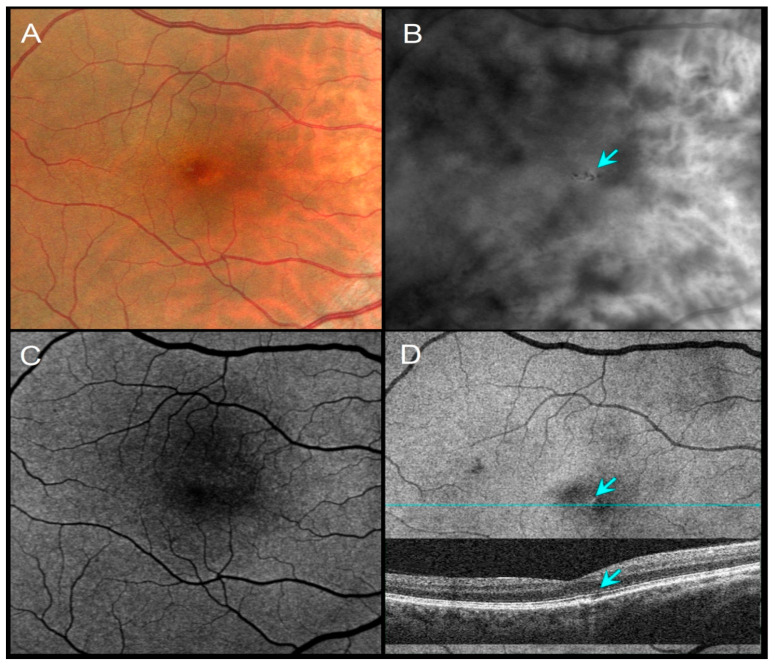
A 51-year-old female with a family history of AMD, VA 20/20. (**A**) CFI shows barely visible pigment migration. (**B**) RAR demonstrates thin dark contrast dots (blue arrow), consistent with small pigment clumps. (**C**) FAF shows no obvious changes in autofluorescence. (**D**) En face OCT (retinal IS/OS slab) and corresponding B-scan showing a thin hypotransmission defect beneath the lesion (blue arrows). Abbreviations: AMD, age-related macular degeneration; CFI, color fundus image; FAF, fundus autofluorescence; OCT, optical coherence tomography; RAR, ring aperture Retro mode; VA, visual acuity.

**Figure 2 diagnostics-16-00042-f002:**
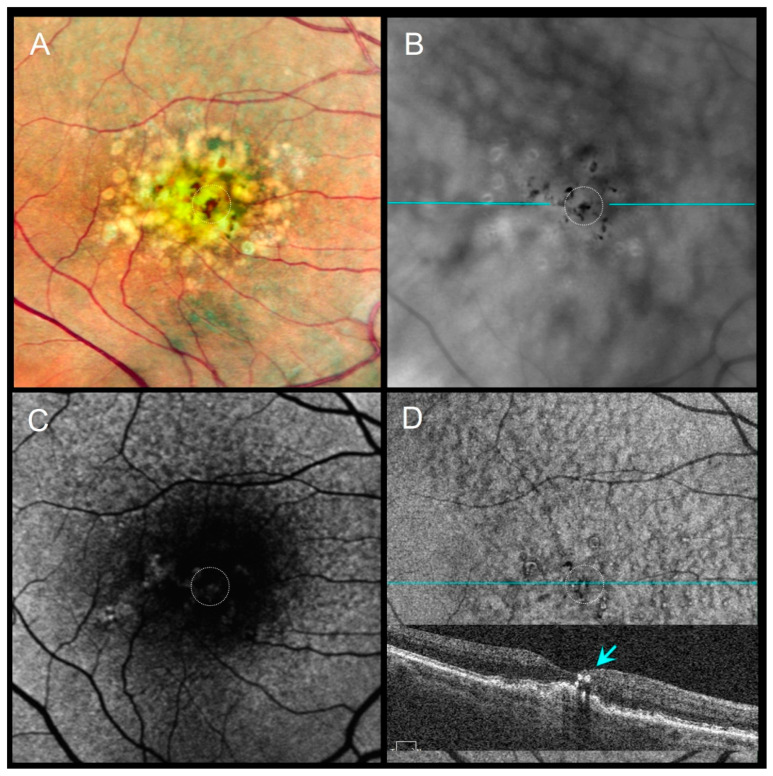
An 86-year-old female with pigment migration over a drusenoid PED, VA 20/63. (**A**) CFI shows pigment migration (dotted circle). (**B**) RAR reveals several mid-sized dark contrast halos corresponding to pigment clumps (dotted circle). (**C**) FAF shows a faint signal change partially obscured by macular pigment. (**D**) En face OCT (retinal IS/OS slab) and corresponding B-scan (blue line on panel (**B**)) show intraretinal HRF overlying the drusenoid PED (dotted circle and blue arrow). Abbreviations: CFI, color fundus image; FAF, fundus autofluorescence; HRF, hyperreflective foci; OCT, optical coherence tomography; PED, pigment epithelial detachment; RAR, ring aperture Retro mode; VA, visual acuity.

**Figure 3 diagnostics-16-00042-f003:**
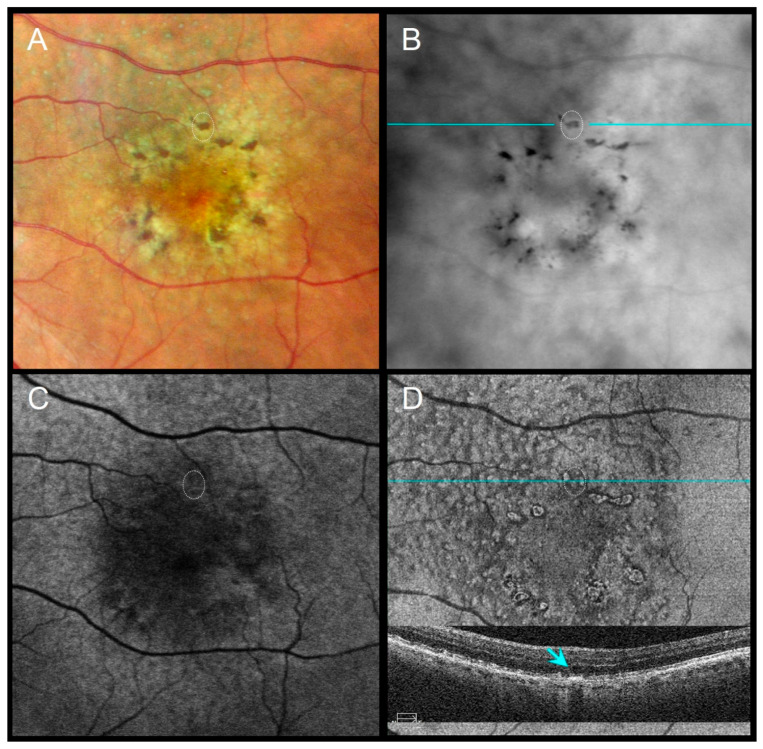
An 85-year-old male with subretinal drusenoid deposits and perifoveal pigment migration sparing the fovea, VA 20/40. (**A**) CFI shows coarse dark lines/dots corresponding to perifoveal pigment migration. (**B**) RAR highlights broad perifoveal dark contrast halos consistent with pigment accumulation. (**C**) FAF displays subtle changes in autofluorescence. (**D**) En face OCT (retinal IS/OS slab) and corresponding B-scan (scan line indicated in panel (**B**)) show thickened RPE (tRPE) beneath areas of pigment migration (blue arrow). Abbreviations: CFI, color fundus image; FAF, fundus autofluorescence; OCT, optical coherence tomography; RAR, ring aperture Retro mode; tRPE, thickened retinal pigment epithelium; VA, visual acuity.

**Figure 4 diagnostics-16-00042-f004:**
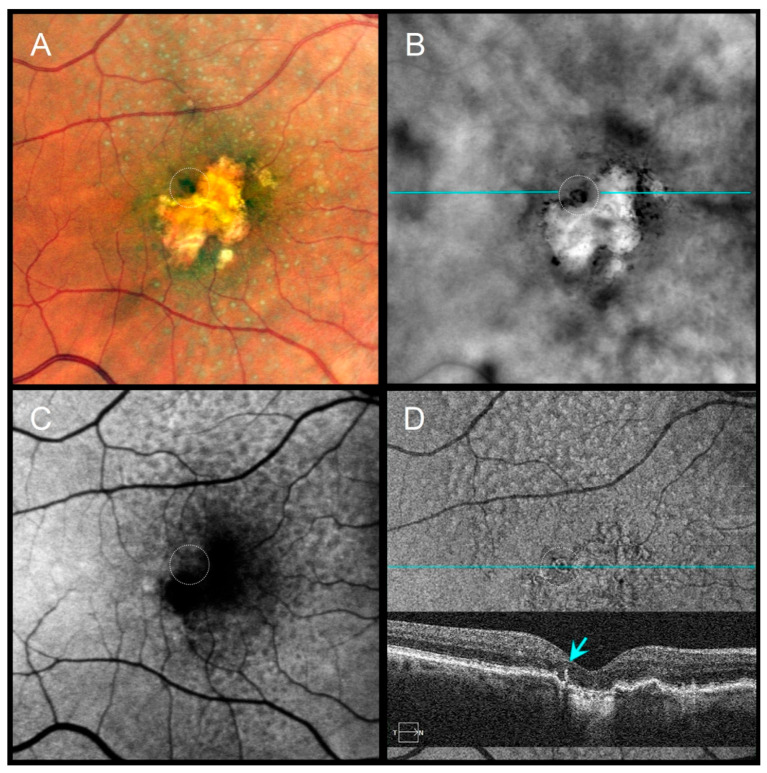
A 79-year-old female with pseudodrusen, geographic atrophy (GA), and a refractile drusen at the superior atrophy margin, VA 20/100. (**A**) CFI shows subtle pigment migration (dotted circle). (**B**) RAR displays a ring-shaped dark contrast halo with a bright central core (dotted circle), consistent with a refractile drusen. (**C**) FAF shows localized signal alterations partially masked by macular pigment. (**D**) En face OCT (retinal IS/OS slab) and corresponding B-scan (scan line indicated in panel (**B**)) demonstrate a hyperreflective ring with a central hypertransmission (blue arrow), compatible with a refractile drusen adjacent to GA. Abbreviations: CFI, color fundus image; FAF, fundus autofluorescence; GA, geographic atrophy; OCT, optical coherence tomography; RAR, ring aperture Retro mode; VA, visual acuity.

**Figure 5 diagnostics-16-00042-f005:**
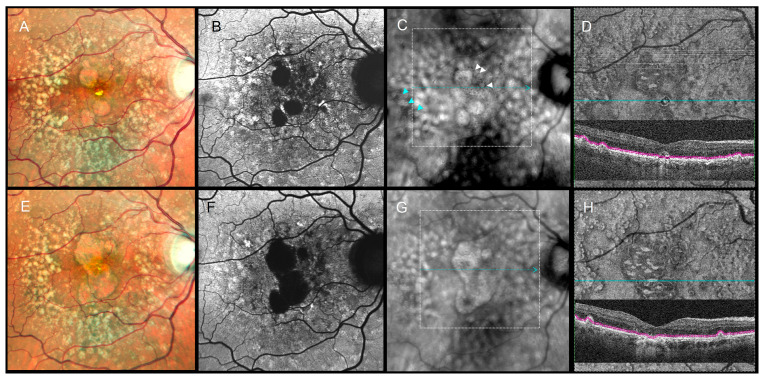
Longitudinal multimodal imaging of a 66-year-old male (2022–2024). Top row (September 2022), VA 20/100: (**A**) CFI, (**B**) FAF, (**C**) RAR, and (**D**) en face OCT (retinal IS/OS slab) with enlarged corresponding B-scan show early subfoveal pigment migration, including a refractile drusen. In panel (**C**), white arrowheads indicate thin pigment migration that was detectable on RAR in 2022. In addition to soft drusen, cuticular drusen at the temporal posterior pole appear on RAR as small clusters with central dark halos and peripheral light rings (blue arrowheads in (**C**)). The white dotted square in (**C**) marks the region shown in the en face OCT slab and B-scan in (**D**). Bottom row (September 2024), VA 20/250: (**E**–**H**) demonstrate progression to central geographic atrophy. The pigment migration previously highlighted by white arrowheads in panel (**C**) preceded subsequent foveal atrophy. In addition to soft drusen, cuticular drusen at the temporal posterior pole appear on RAR as small clusters with central dark halos and peripheral light rings (blue arrowheads in (**C**)). The white dotted square in (**G**) again indicates the area displayed in the en face OCT slab and B-scan in (**H**). Abbreviations: CFI, color fundus image; FAF, fundus autofluorescence; GA, geographic atrophy; OCT, optical coherence tomography; RAR, ring aperture Retro mode; VA, visual acuity.

**Table 1 diagnostics-16-00042-t001:** Contingency tables comparing RAR with en face OCT (reference standard) and with CFIs for the detection of pigment migration in 80 eyes (abbreviations: CFI, color fundus image; OCT, optical coherence tomography; RAR, ring aperture Retro mode). Note: low TN counts reflect enrichment for pigment migration and limit specificity estimation. Sensitivity/PPV/accuracy CIs: Wilson 95% CI. F1 CI: patient-level cluster bootstrap resampling.

	En Face OCT (Reference)	CFIs
True positives (TPs)	71	43
False negatives (FNs)	4	4
False positives (FPs)	5	33
True negatives (TNs)	0	0
Total	80	80

**Table 2 diagnostics-16-00042-t002:** Performance metrics of RAR compared with en face OCT and CFIs, including sensitivity, specificity, positive predictive value (PPV), accuracy, and F1 score (abbreviations: CFI, color fundus image; OCT, optical coherence tomography; RAR, ring aperture Retro mode). * Specificity not estimable due to the absence of true negatives in this enriched cohort. Confidence intervals for sensitivity and PPV were calculated using Wilson’s method; F1 score and kappa confidence intervals were estimated via patient-level cluster bootstrap resampling to account for inter-eye correlation (N/A: non applicable).

Metric	vs. En Face OCT	vs. CFIs
Sensitivity (%)	94.7 (95% CI 87.1–97.9)	91.5 (95% CI 80.1–96.6)
Specificity (%)	N/A *	N/A *
Positive predictive value (%)	93.4 (95% CI 85.5–97.2)	56.6 (95% CI 45.4–67.1)
Accuracy (%)	88.8 (95% CI 80.0–94.0)	53.8 (95% CI 42.9–64.3)
F1 score (%)	94.0 (95% CI 89.7–97.4)	69.9 (95% CI 59.6–78.8)

**Table 3 diagnostics-16-00042-t003:** Characteristic morphologic patterns of pigment migration displayed on color fundus images (CFIs), en face OCT, B-scan OCT, and ring aperture Retro mode (RAR) imaging. In RAR, dark halos denote hyporeflectivity and bright halos denote hyperreflectivity, consistent with its indirect transillumination principle. As lesion appearance on en face OCT is known to vary with the depth and thickness of the selected slab, the descriptions provided here reflect the features observed using the two standard Cirrus preset slabs applied in our study; customized slabs may yield different patterns, as previously reported. Imaging characteristics were adapted from established descriptions in the literature [3,11,17,18,21,22].

Lesion Type	CFI	En Face OCT	B-Scan OCT	RAR
**Pigment migration/HRF**	Small dark clumps or spicules; clustered	Bright focal dots [3]; Dark foci with shadowing [3]	Hyperreflective intraretinal foci [3]	Granular hyperreflective dots [18,22]
**tRPE**	Thick, irregular, dark areas	Edge signals; HypoTD [3]	Thickened/irregular RPE; reduced transmission [3]	Plateau-like dark halos [21,22]
**Early atrophy/nascent GA**	Pale areas	Loss of normal pattern; HyperTD patches [3]	RPE attenuation + increased choroidal signal	Light homogeneous patches [17]
**Refractile drusen**	Crystalline shiny points	Bright ring/dot; HyperTD [11]	Calcified drusen with reflective core [11]	Ring-like hyperreflectivity (bright halos) [17,18,21]
**Cuticular drusen**	Multiple small yellow spots	Clustered bright dots; small ring-like HypoTD [11]	Small dome-shaped RPE elevations	Small clustered hyperreflective dots [17,21]

## Data Availability

The data presented in this study are available on request from the corresponding author due to privacy concerns.

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
