# Peer review of "Diagnostic Performance of Ring Aperture Retro Mode Imaging for Detecting Pigment Migration in Age-Related Macular Degeneration"

_diagnostics, 2025, doi:10.3390/diagnostics16010042_

Round 1

Reviewer 1 Report

Comments and Suggestions for Authors

Dear editor

Thank you for the opportunity to review this manuscript.

This retrospective single-center study evaluates the diagnostic performance of ring aperture Retro mode (RAR) infrared imaging for detecting pigment migration in age-related macular degeneration (AMD) against a reference standard of en face OCT corroborated by B-scans. In 80 eyes from 61 patients, RAR achieved a reported sensitivity of 94.7% and PPV of 93.4% relative to en face OCT, frequently outperforming color fundus imaging (CFI) and, in selected situations, fundus autofluorescence (FAF).

The paper provides, to my knowledge, the first focused, systematic evaluation of the ring aperture configuration of retroillumination (RAR) for pigment migration in AMD, distinct from the more commonly used deviated apertures (DL/DR).  However, I noted several issues that should be addressed before the publication of the manuscript.

1- The choice of en face OCT as the singular reference standard, although pragmatic, may not capture all pigment redistribution detectable by RAR; some RAR “false positives” may reflect true pigment changes outside the en face slabs or below OCT’s sensitivity, potentially biasing performance metrics.

2- Did you explore additional en face OCT slabs or customized slab thicknesses, and if so, how did this affect agreement with RAR? If not, can you comment on how slab selection might influence your conclusions?

3- For the four longitudinal GA cases, can you quantify the lead time from RAR-detected pigmentation to OCT/FAF-confirmed foveal atrophy and describe the specific pigment patterns predictive of progression?

4- Inclusion criteria enriched for eyes with pigment migration on at least one modality, leading to a highly positive dataset; this undermines specificity/NPV estimation and inflates PPV due to high prevalence. Can you justify the impact of this spectrum bias on specificity/NPV, and consider a sensitivity analysis including a broader unselected AMD cohort?

5- Zero true negatives against OCT (TN=0) and the resulting specificity=0% indicate a spectrum bias in the sample and limit interpretability of specificity/NPV.

6- The near-perfect intergrader agreement for RAR (κ=1.0) is striking; while plausible if RAR contrast is strong, confidence intervals and an explicit grading protocol description (e.g., training, adjudication) would increase confidence.

7- Confidence intervals and uncertainty estimates for sensitivity, PPV, and kappa are not provided; no power or sample size considerations are discussed. Please report 95% confidence intervals for sensitivity, PPV, kappa values, and any other key metrics. Lack of 95% confidence intervals for sensitivity/PPV, and absence of statistical adjustment for within-patient correlation, reduce inferential strength. A generalized estimating equation or cluster bootstrap would be appropriate.

8- Lack of adjustment for eye-level clustering (80 eyes from 61 patients) may overstate precision in agreement and diagnostic performance estimates.

9- How were graders masked across modalities during the reading sessions to avoid cross-modality recall bias? Were readings randomized by modality and eye?

10- Could you elaborate on the acquisition protocol standardization for RAR (e.g., exposure/flash settings) and provide reproducibility data across technicians/operators or within the same eye over repeated acquisitions?

11- All images were acquired by a single experienced operator, potentially limiting generalizability to routine clinical acquisition conditions. Only a single-site/single-operator dataset is presented; device-specific performance, operator dependence (e.g., flash-intensity optimization), and external validity remain untested.

12- Performance of FAF is qualitatively discussed but not fully quantified against the reference (e.g., FAF vs OCT contingency and metrics are not tabulated), and no subgroup analysis by AMD stage is presented. Can you provide full contingency tables and diagnostic performance metrics for FAF vs the reference standard, and subgroup performance for iAMD, GA, and nAMD? Subgroup analyses by AMD stage (iAMD, GA, nAMD), media opacity status, and foveal vs extrafoveal location would make the findings more actionable.

13- A more explicit discussion of near-infrared autofluorescence (NIR-AF) and its potential role in pigment detection would strengthen the contextualization of RAR.

14- The paper could better integrate recent insights on en face OCT slab physics and artifacts to contextualize the reference standard’s limitations.

Author Response

We sincerely thank the Reviewer for the thorough and insightful evaluation of our manuscript. We appreciate the time devoted to this detailed review and the constructive comments, which have helped us substantially improve the methodological clarity, statistical rigor, and contextual interpretation of our study. We have carefully addressed each point raised below, and the manuscript has been revised accordingly.

Comment 1:

The choice of en face OCT as the singular reference standard, although pragmatic, may not capture all pigment redistribution detectable by RAR; some RAR “false positives” may reflect true pigment changes outside the en face slabs or below OCT’s sensitivity, potentially biasing performance metrics.

Response:
We agree with this important observation. En face OCT was selected as a pragmatic structural reference standard because it allows precise localization and confirmation of pigment-related changes through mandatory B-scan correlation. However, we acknowledge that some pigment redistribution detected by RAR may occur outside the selected en face slabs or below the sensitivity of OCT.

We have now explicitly discussed this limitation in both the Methods (reference standard definition) and the Discussion, emphasizing that RAR “false positives” relative to OCT may represent biologically relevant pigment changes not fully captured by slab-based OCT analysis. This contextualization has been added to clarify interpretation of diagnostic performance metrics.

Comment 2:

Did you explore additional en face OCT slabs or customized slab thicknesses, and if so, how did this affect agreement with RAR? If not, can you comment on how slab selection might influence your conclusions?

Response:
We thank the reviewer for highlighting this key technical point. In this study, we intentionally restricted en face OCT analysis to the two standard Cirrus preset slabs (ellipsoid zone and sub-RPE/choroidal slabs) to ensure consistency across all eyes and reproducibility in routine clinical practice.

We now explicitly state in the Methods and Table 3 legend that lesion appearance on en face OCT is slab-dependent and may vary with customized slab depth or thickness, as previously reported. This point is further discussed in the Discussion, where we acknowledge that alternative slab configurations could potentially increase agreement with RAR and represent an important direction for future work.

Comment 3:

For the four longitudinal GA cases, can you quantify the lead time from RAR-detected pigmentation to OCT/FAF-confirmed foveal atrophy and describe the specific pigment patterns predictive of progression?

Response:
We have clarified this point in the Results and Discussion. In the four GA eyes, pigment migration detected on RAR preceded foveal involvement by at least six months before confirmation by OCT and FAF. The observed patterns primarily consisted of thin or granular pigment redistribution at the margins of evolving atrophy.

Given the small number of longitudinal cases, we refrained from formal predictive modeling but explicitly describe these observations as hypothesis-generating and indicative of potential prognostic relevance.

Comments 4 & 5

Inclusion criteria enriched for eyes with pigment migration on at least one modality, leading to a highly positive dataset; this undermines specificity/NPV estimation and inflates PPV due to high prevalence. Can you justify the impact of this spectrum bias on specificity/NPV, and consider a sensitivity analysis including a broader unselected AMD cohort?

Zero true negatives against OCT (TN=0) and the resulting specificity=0% indicate a spectrum bias in the sample and limit interpretability of specificity/NPV.

Response:
We fully agree. The study design intentionally enriched for eyes with pigment migration in order to compare detection performance across imaging modalities, rather than to estimate population-level prevalence or screening performance.

We now explicitly state this enrichment in the Methods; clarify in the Results that specificity and NPV are non-informative in this dataset due to very low true-negative counts; emphasize in the Discussion that PPV and sensitivity are the most meaningful metrics under these conditions. This limitation is now clearly acknowledged and appropriately contextualized.

Comment 6

Near-perfect intergrader agreement for RAR (κ = 1.0) is striking; confidence intervals and grading protocol description would help.

Response:
We appreciate this comment. We have now added 95% confidence intervals for all κ statistics and clarified the grading protocol in the Image Analysis section (independent reading, masking between graders, separate reading sessions by modality, and consensus adjudication)

We note that the high κ value for RAR likely reflects its strong contrast and intuitive visualization, but we agree that such results should be interpreted cautiously and in context.

Comments 7 & 8

Confidence intervals and uncertainty estimates for sensitivity, PPV, and kappa are not provided; no power or sample size considerations are discussed. Please report 95% confidence intervals for sensitivity, PPV, kappa values, and any other key metrics. Lack of 95% confidence intervals for sensitivity/PPV, and absence of statistical adjustment for within-patient correlation, reduce inferential strength. A generalized estimating equation or cluster bootstrap would be appropriate.

Lack of adjustment for eye-level clustering (80 eyes from 61 patients) may overstate precision in agreement and diagnostic performance estimates.

Response:
This is an important point, and we have substantially strengthened the statistical analysis accordingly.

We now: report 95% confidence intervals for sensitivity, PPV, F1 score, and κ statistics. We use a patient-level cluster bootstrap approach, resampling patients (and all corresponding eyes) to account for intra-subject correlation. This approach is described in detail in the Statistical Analysis section and directly addresses potential overestimation of precision due to eye-level clustering.

Comment 9

How were graders masked across modalities during the reading sessions to avoid cross-modality recall bias? Were readings randomized by modality and eye?

Response:
We have clarified this in the Image Analysis section. Images from different imaging modalities were reviewed in separate reading sessions, and graders were masked to each other’s findings. This procedural clarification has been added to minimize concerns regarding cross-modality recall bias.

Comments 10 & 11

Could you elaborate on the acquisition protocol standardization for RAR (e.g., exposure/flash settings) and provide reproducibility data across technicians/operators or within the same eye over repeated acquisitions?

All images were acquired by a single experienced operator, potentially limiting generalizability to routine clinical acquisition conditions. Only a single-site/single-operator dataset is presented; device-specific performance, operator dependence (e.g., flash-intensity optimization), and external validity remain untested.

Response:
We agree that acquisition parameters are critical for RAR imaging. All images were acquired by a single experienced operator to ensure consistent optimization of illumination settings, particularly flash intensity, which is essential for meaningful RAR interpretation.

We now explicitly discuss in the Discussion that this choice enhances internal consistency but may limit generalizability. RAR likely has a learning curve, similar to that previously reported for DR DR Retro mode imaging.

Multi-operator and multi-center validation might be useful in future studies.

Comment 12

Performance of FAF is qualitatively discussed but not fully quantified against the reference (e.g., FAF vs OCT contingency and metrics are not tabulated), and no subgroup analysis by AMD stage is presented. Can you provide full contingency tables and diagnostic performance metrics for FAF vs the reference standard, and subgroup performance for iAMD, GA, and nAMD? Subgroup analyses by AMD stage (iAMD, GA, nAMD), media opacity status, and foveal vs extrafoveal location would make the findings more actionable.

Response:
FAF performance is now more explicitly contextualized. We clarify in the Results that FAF was qualitatively assessed but not emphasized as a binary comparator due to known variability related to macular pigment absorption, fluorophore composition, and disease stage.

Given the enriched design and limited true-negative counts, extensive subgroup analyses would risk overinterpretation. We therefore acknowledge this as a limitation and emphasize the complementary, rather than competitive, role of FAF.

Comment 13

A more explicit discussion of near-infrared autofluorescence (NIR-AF) and its potential role in pigment detection would strengthen the contextualization of RAR.

Response:
We have expanded the Discussion to explicitly address NIR-AF, highlighting its relevance to melanin and melanolipofuscin detection and its conceptual overlap with RAR. We clarify that while NIR-AF and RAR both rely on infrared light, they probe different physical principles and provide complementary information.

Comment 14

The paper could better integrate recent insights on en face OCT slab physics and artifacts to contextualize the reference standard’s limitations.

Response:
We have added a dedicated paragraph in the Discussion addressing slab-dependent artifacts in en face OCT (projection effects, mirror artifacts, shadowing), and we emphasize the importance of mandatory B-scan correlation. This contextualizes the limitations of the reference standard and strengthens interpretation of RAR–OCT discrepancies.

Reviewer 2 Report

Comments and Suggestions for Authors

The manuscript is interesting and deals with an important topic, however, it needs improvement.

  1. Details of the ethical approval should be included in the text
  2. The minimum sample size to achieve the confidence level of 95% should be calculated in the Statistical Analysis section
  3. The study included 80 eyes of the 61 patients, which is against Hoffer recommendation (this should be added in the limitations of the study)
  4. The results should be discussed in subgroups according to gender
  5. According to the Authors, does pigment migration in AMD differ between men and women, and if so, why?
  6. The clinical value of the study should be emphasized in the Conclusions section

Author Response

We sincerely thank the Reviewer for the careful evaluation of our manuscript and for the constructive suggestions. We appreciate the positive assessment of the scientific relevance of our work and have addressed each comment in detail below. The manuscript has been revised accordingly to improve clarity, methodological transparency, and clinical relevance.

Comment 1

Details of the ethical approval should be included in the text.

Response:
Thank you for this comment. We have clarified the ethical approval details in the Materials and Methods section. The manuscript now explicitly states that the study was approved by the Ethical Committee of the French Society of Ophthalmology (SFO) (IRB 00008855 SFO IRB#1), conducted in accordance with the Declaration of Helsinki, and that written informed consent was obtained from all participants.

Comment 2

The minimum sample size to achieve the confidence level of 95% should be calculated in the Statistical Analysis section.

Response:
We appreciate this important point. Given the retrospective and exploratory nature of the study, which was designed to compare diagnostic detectability across imaging modalities rather than to test a predefined hypothesis, a formal a priori sample size calculation was not performed.

To address statistical uncertainty, we now explicitly report 95% confidence intervals for all key performance metrics (κ statistics, sensitivity, PPV, and F1 score). These intervals were derived using a patient-level cluster bootstrap approach to account for intra-subject correlation. This strategy provides a robust assessment of estimate precision without relying on assumptions required for prospective power calculations. This clarification has been added to the Statistical Analysis section.

Comment 3

The study included 80 eyes of the 61 patients, which is against Hoffer recommendation (this should be added in the limitations of the study).

Response:
We agree that inclusion of both eyes from the same patient may introduce intra-subject correlation, as highlighted by Hoffer. This limitation is now explicitly acknowledged in the Discussion.

In addition, we addressed its statistical implications by applying patient-level cluster bootstrap resampling in the analysis, which accounts for eye-level clustering and reduces the risk of overestimating precision. These methodological considerations are now clearly described.

Comment 4

The results should be discussed in subgroups according to gender.

Response:
We thank the Reviewer for this suggestion. Gender distribution is now explicitly reported in the Results section. However, the study was not powered to perform reliable gender-stratified analyses of diagnostic performance. Given the modest sample size and exploratory design, such subgroup analyses would risk overinterpretation.

This limitation is now acknowledged in the Discussion, and we identify sex-based analyses as an important objective for future, larger-scale studies.

Comment 5

According to the Authors, does pigment migration in AMD differ between men and women, and if so, why?

Response:
At present, there is no consistent evidence demonstrating sex-specific differences in pigment migration patterns in AMD. While some studies have reported sex-related differences in AMD prevalence or progression, pigment migration appears primarily driven by local retinal factors such as RPE dysfunction, photoreceptor stress, and microenvironmental changes rather than sex-specific biological pathways.

We have added a brief clarification in the Discussion noting that the present study was not designed to address this question and that targeted investigations would be required to explore potential sex-related differences.

Comment 6

The clinical value of the study should be emphasized in the Conclusions section.

Response:
We agree and have revised the Conclusions section accordingly. The revised conclusion now more explicitly emphasizes the clinical relevance of ring aperture Retro mode imaging, particularly its practical advantages for routine monitoring, its utility in early disease stages and near the fovea, and its added value in cases with media opacity. Its complementary role alongside OCT in real-world clinical practice is now clearly highlighted.

Again, we thank the Reviewer for these thoughtful and constructive comments, which have helped strengthen the manuscript. We believe that the revisions have improved methodological transparency, statistical rigor, and clinical contextualization, and we hope that the revised version satisfactorily addresses all concerns raised.

Reviewer 3 Report

Comments and Suggestions for Authors

This is a very interesting retrospective study concerning the use of ring aperture retro mode imaging for detecting pigment migration in Age related macular degeneration.

The topic of this manuscript is really challenging and could be of interest for scientific readers. In fact, the use of multimodal imaging for the diagnosis and management of retinal diseases such as AMD is a very current and modern topic, considering that the right and early diagnosis of this disease could guide a better therapeutic approach, with better outcomes for the patients.

Considering the Abstract, it is fine, very clear and properly summarizes all the key points of the manuscript.

Considering the Introduction section, it is well-written and well-organized, providing all the information needed to better understand of the topic and clearly stating the aim of the study.

The Materials and Methods section is very clear, with no ethical concerns and properly describing all the ophthalmological evaluations performed in this study. Also the statistical analysis is properly described and it is relevant for the nature of the study.

The Results section is well-organized, providing all the main results in a clear way, also with the help of the Tables and Figures, that are crucial to understand the key results of the study and are relevant for Diagnostic Journals.

The Discussion section properly analyzes the published literature on the discussed topic, also comparing this with the results of the present study. Also the limitations of the study are clearly described.

The Conclusion section properly summarizes the key points of the study.

I have just few comments to try to improve this manuscript:

1) Remove the abbreviation AMD in the Title of the manuscript.

2) I suggest the authors to specify better the inclusion and exclusion criteria for this study.

3) I suggest the authors to add the future perspectives on the discussed topic at the end of the manuscript.

Author Response

We sincerely thank Reviewer 3 for the very positive and thoughtful evaluation of our manuscript. We greatly appreciate the reviewer’s careful reading, constructive suggestions, and encouraging comments regarding the clinical relevance, methodological clarity, and presentation of our work.

We have addressed all comments as outlined below.

Comment 1

Remove the abbreviation AMD in the Title of the manuscript.

Response:
We thank the reviewer for this suggestion. The title has been revised to remove the abbreviation “AMD,” and the full term “age-related macular degeneration” is now used for clarity and consistency with journal style.

Comment 2

Please specify better the inclusion and exclusion criteria for this study.

Response:
We agree that clearer specification of inclusion and exclusion criteria improves transparency. The Patients subsection of the Materials & Methods has been revised to explicitly detail eligibility criteria, including diagnostic requirements, imaging completeness, and image quality standards, as well as exclusion criteria.

Comment 3

Please add future perspectives on the discussed topic at the end of the manuscript.

Response:
We appreciate this helpful suggestion. A dedicated paragraph addressing future perspectives has been added to the Discussion, highlighting the potential role of longitudinal RAR imaging, multimodal integration (including OCT and near-infrared autofluorescence), and also the need for multicenter validation studies.

Round 2

Reviewer 1 Report

Comments and Suggestions for Authors

Dear Editor

Thank you for providing me with the opportunity to review this manuscript.

The authors have thoroughly addressed all my concerns. The revisions are comprehensive and responsive, aligning well with the comments. The manuscript is now stronger, with enhanced scientific and editorial quality.

I have no further comments; therefore, I recommend the publication of the manuscript.

Best wishes

Reviewer 2 Report

Comments and Suggestions for Authors

The manuscript has been revised sufficiently.